# Characteristics and Formation Mechanisms of Spring SST Anomalies in the South China Sea and Its Adjacent Regions

**Wenjuan Gao [1], Song Yang [2,3,4], Xiaoming Hu [2,3,4,*] , Wei Wei [2,3,4] and Yanglin Xiao [1]**

[1]  China Meteorological Administration Training Centre Hunan Branch, Changsha 410125, China;
    gaowj5@mail2.sysu.edu.cn (W.G.); zenglingdong2014@163.com (Y.X.)
[2]  School of Atmospheric Sciences, Sun Yat-sen University, Guangzhou 510275, China;
    yangsong3@mail.sysu.edu.cn (S.Y.); weiwei48@mail.sysu.edu.cn (W.W.)
[3]  Guangdong Province Key Laboratory for Climate Change and Natural Disaster Studies,
    Sun Yat-sen University, Guangzhou 510275, China
[4]  Southern Marine Science and Engineering Guangdong Laboratory (Zhuhai), Zhuhai 519000, China
*   Correspondence: huxm6@mail.sysu.edu.cn; Tel.: +86-020-8411-1286

**Abstract:** Characteristics of the springtime sea surface temperature anomalies (SSTAs) in the South China Sea and its adjacent regions (SCSAR), as well as their possible impacts on the Asian and Indo-Pacific climate, were investigated by using multiple datasets. According to the result from an empirical orthogonal function (EOF) analysis on the spring SSTAs in the SCSAR, the dominant pattern is a uniformly warming pattern in the whole SCSAR region. While the second mode is a sandwich pattern with cold SSTA over the central SCSAR centered near 10° N, flanked by warm SSTA over the northern oceans near 25° N and in the subtropics near 10° S. The uniformly warming pattern is associated with the anomalous warming in the Indian Ocean from the preceding autumn to the spring, and the sandwich pattern is mainly caused by the El Niño-Southern Oscillation. In the uniformly warming pattern, rainfall increases in the Meiyu region and decreases over the southern South China Sea (SCS). In the sandwich pattern, the anomalous anticyclone at 850-hPa causes less rainfall in the Philippine Sea, the Marine Continent, and the SCS. The positive rainfall anomalies could be found in the northern SCS and adjacent regions. Associated with the second EOF mode, there is a wave train emitted from the SCSAR to East Asia, northwest Pacific, and North America. The wave train spreads the energy from mid-latitudes to higher latitudes through atmospheric teleconnection, which can even influence the North American atmospheric circulation in spring.

**Keywords:** South China Sea and surrounding areas; spring SSTAs; interannual variability; climate effect

## 1. Introduction

The South China Sea (SCS) and its adjacent region (SCSAR) consist of the SCS, the Maritime Continent, and the Northwest Pacific. It is a key region for the Asian summer monsoon [1–6]. The SCS is the upstream region of the East Asian summer monsoon [7,8]. Abundant water vapor and heat are transported from the SCSAR to East Asia, affecting the temperature and rainfall variations over East Asia and the Northwest Pacific. Besides, the SCSAR is adjacent to the equatorial warm pool, where located the rising branch of the Hadley and the Walker circulations. Thus, the SCSAR is also a key region for the atmospheric teleconnection patterns over the Indo-Pacific Ocean and East Asia, and thorough investigations of the climate variations in SCSAR are helpful for better understanding the key role of the SCS in East Asian climate variability.

The sea surface temperature anomalies (SSTAs) in the SCS exhibits obvious interannual variability. Many studies have focused on the SCS SSTAs in the summer and winter seasons. It is also known that the summer SSTAs in the SCS are sensitive to El Niño-Southern Oscillation (ENSO) events [9], and the cold tongue of the SCS in winter also has a close correlation with the Nino 3 index [10]. It is also known that the seasonal characteristics of the SCS circulations are related to the El Niño events [11,12]. The SSTAs in the eastern equatorial Pacific influence the anticyclones in the SCS and the Northwest Pacific [13]. It also affects the precipitation variability in China by modulating the Walker circulation and local Hadley circulation over the tropical Pacific Ocean [14,15]. However, the characteristics of the SSTAs in the SCSAR during spring are seldom investigated.

During boreal spring, the East Asian monsoon system establishes rapidly, and the El Niño or La Niña events weaken or terminate [16,17]. Due to the rapid evolutions of both the Asian monsoon and ENSO, the SSTA signals in the SCS are weak and unstable [18]. The characteristics of spring SSTA on multiple timescales have not been fully understood. Previous studies have found that the warming trend of SST over the Indo-Pacific warm pool played an essential role in the variation of the Hadley circulation during the boreal spring [19]. In addition, colder (warmer) SSTs over the western Pacific and inactive (active) convection over the Southern Philippines suppress (favor) the Northwestward development of the South Asian high in late April. As a result, the entire Asian summer monsoon onset process is later (earlier) than normal [17]. Min et al. [20] pointed out that due to the influences of the Indian monsoon and the East Asian monsoon, the SSTAs in the SCS, the Bay of Bengal, and the Arabian Sea mostly exhibit a zonally homogenous distribution.

When the SSTAs are positive (negative) in spring, the western Pacific subtropical high appears strong (weak) in summer, and its position leans toward the west (east), resulting in more (less) rainfall in the Yangtze River and less (more) rainfall in North and South China [20]. However, the impacts of the spring SSTAs in the SCS on the Asian and Indo-Pacific climate are still unclear. This study is to explore the interannual variability of spring SSTAs in the SCSAR and its potential impacts on spring climate variations over the Asian and Indo-Pacific.

The rest of this paper is organized as follows. Section 2 described the data and methods. Section 3 discussed the main characteristics and formation mechanisms of spring SSTAs in the SCSAR, as well as their possible impacts on the Asian and Indo-Pacific climate. Finally, Section 4 presented the conclusions and further discussions.

## 2. Data and Methods

### 2.1. Data

This study mainly used the monthly data from the National Centers for Environmental Prediction/National Center for Atmospheric Research (NCEP/NCAR) reanalysis product. It includes the zonal and meridional winds files and the geopotential height files with a 2.5° × 2.5° horizontal resolution and 17 layers in the vertical direction [21]. The Extended Reconstructed Sea Surface Temperature (ERSST) version 3 was applied. It is supported by the National Oceanic and Atmospheric Administration (NOAA) and has a horizontal resolution of 2° × 2° [22,23]. The NOAA's Precipitation Reconstruction (PREC) data that have a resolution of 2.5° × 2.5° were also applied [24]. All the above fields are available for the analysis period of 1948–2018. The Optimum Interpolation Sea Surface Temperature (OISST) version 2 supported by NOAA, with a horizontal resolution of 1° × 1°, are also used for the analysis period of 1982–2018 [25].

### 2.2. Statistical Methods

The main methodologies included empirical orthogonal function (EOF) decomposition, composite analysis, linear regression analysis, and correlation analysis. We focused on the boreal spring (March-April-May: MAM) SSTAs of the SCSAR on interannual scales. And discussed their possible impacts on the Asian and Indo-Pacific climate in spring, summer (June-July-August: JJA), and autumn

(September-October-November: SON). All variables' anomalies were calculated by subtracting the climatological annual cycle from the original monthly SST data (the climatological reference period is from 1948 to 2018). We also removed the long-term trend to focus on the interannual timescales. The wave activity flux calculated in this paper is developed by Takaya and Nakamura [26], namely TN wave activity flux.

The analysis domain is 100°–140° E/10° S–30° N, selected based on the largest correlation coefficient between region-averaged SSTA indices and grid-point SSTAs (the rectangle in Figure 1). This area is defined as the South China Sea and its adjacent regions (SCSAR).

## 3. Results

### 3.1. Main Distribution Patterns of MAM SCSAR SSTAs

In order to represent the primary characteristic of the SCSAR SSTAs on interannual timescales, the SCSAR index (SCSARI, Figure 1b) is defined as the standardized time series of the domain average SSTAs in MAM in the SCSAR (100°–140° E/10° S–30° N). Figure 1a shows the spatial pattern of regression of the MAM SSTAs with SCSARI, with a uniformly warm anomaly in the whole SCSAR and large positive value concentrated in the coastal areas of Southeast Asia, the northern SCS, and the Northwest Pacific Ocean.

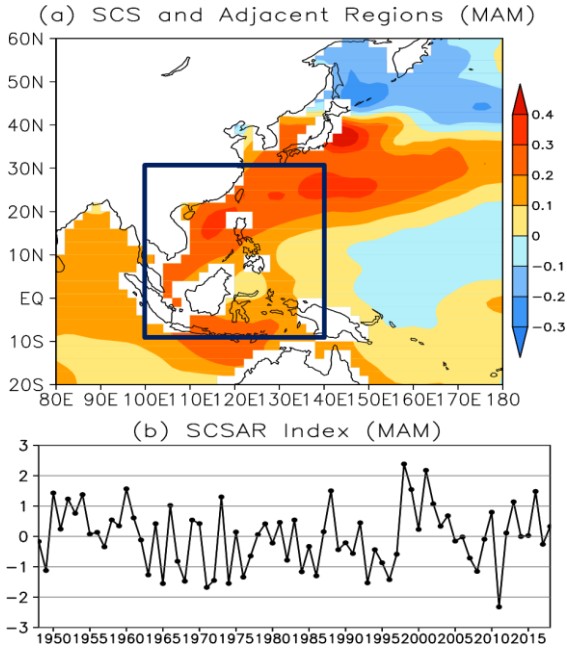

**Figure 1.** (**a**) Regression of the MAM sea surface temperature anomalies (SSTAs) (shadings, units: °C) with (**b**) South China Sea and its adjacent regions (SCSAR) index. (**b**) Standardized temporal evolution of SSTAs in spring from 1949 to 2018 in the SCSAR (100°–140° E/10° S–30° N), which is defined as SCSAR index. The long-term trend of the SCSAR index has been removed. The rectangle in (**a**) marks the SCSAR over which SSTAs are averaged to calculate the SCSAR index in (**b**).

The result of EOF analysis on MAM SSTAs in the SCSAR (Figure 2) shows that the cumulative variance contributions of the first two modes reach 61.0%, both exceeding the eigenvalue significance test proposed by North et al. [27]. For the first mode, the MAM SSTAs show a uniformly warming pattern, with a variance contribution of 45.15%. The warm center is located along the western Pacific coast in the northern SCS. For the second mode, the MAM SSTAs present a sandwich pattern with cold SSTAs over the central SCSAR centered near 10° N, flanked by warm SSTAs to its north and south, with warming centers at about 25° N and 10° S, respectively. The variance contribution of the second mode is 15.83%.

The correlation coefficient between the first mode time series (PC1) and SCSARI is as high as 0.98. The SCSARI is selected as the index to reflect the main characteristics of MAM SSTAs in the SCSAR.

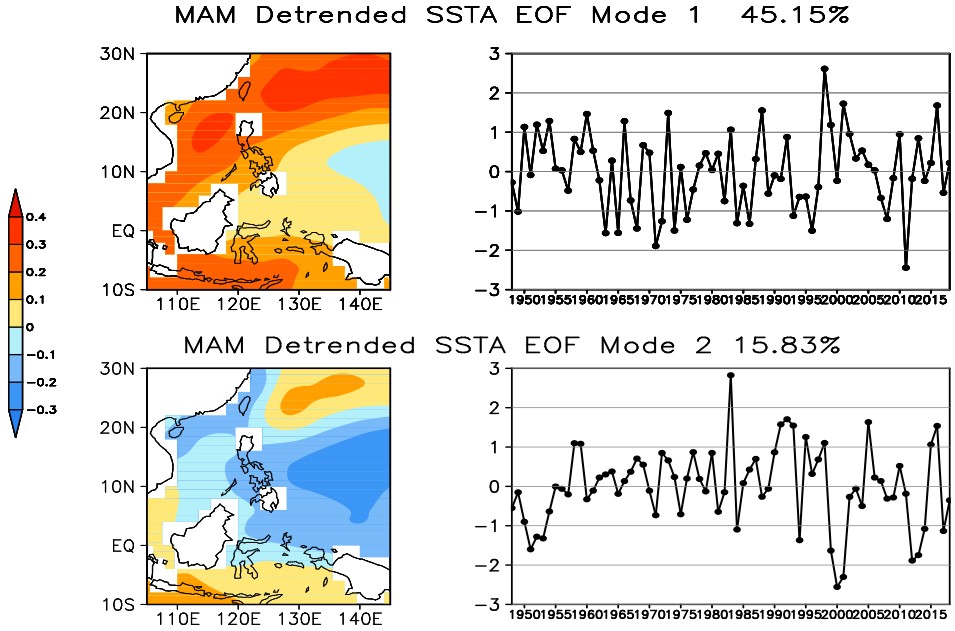

**Figure 2.** Spatial distributions of the first two modes of empirical orthogonal function (EOF) analysis for MAM SSTAs. The time series has been standardized, and the corresponding SSTA (units: °C) distribution shows the regression of the time series.

### 3.2. Analysis of the Characteristics and Formation Mechanisms of MAM SSTAs

To explore how SSTAs and precipitation anomalies in the Asian and Indo-Pacific regions are related to the uniformly warming in the SCSAR in spring, regression analyses are conducted between SCSARI and various physical fields (SST, rainfall, winds at 850-hPa and geopotential height at 500-hPa) in spring (Figure 3). The equatorial Indian Ocean, the Bay of Bengal, and the SCSAR show positive SSTAs (Figure 3a). In the Northwest Pacific, there is a warm core extending from the east coast of China to the mid-latitudes. We define it as the Northwest Pacific warm tongue where is located around 30° N. The equatorial central Pacific Ocean also shows warm anomalies, while cold water anomalies surrounding the "warm tongue" appear in the North Pacific, which yields a flipped-C-shaped distribution.

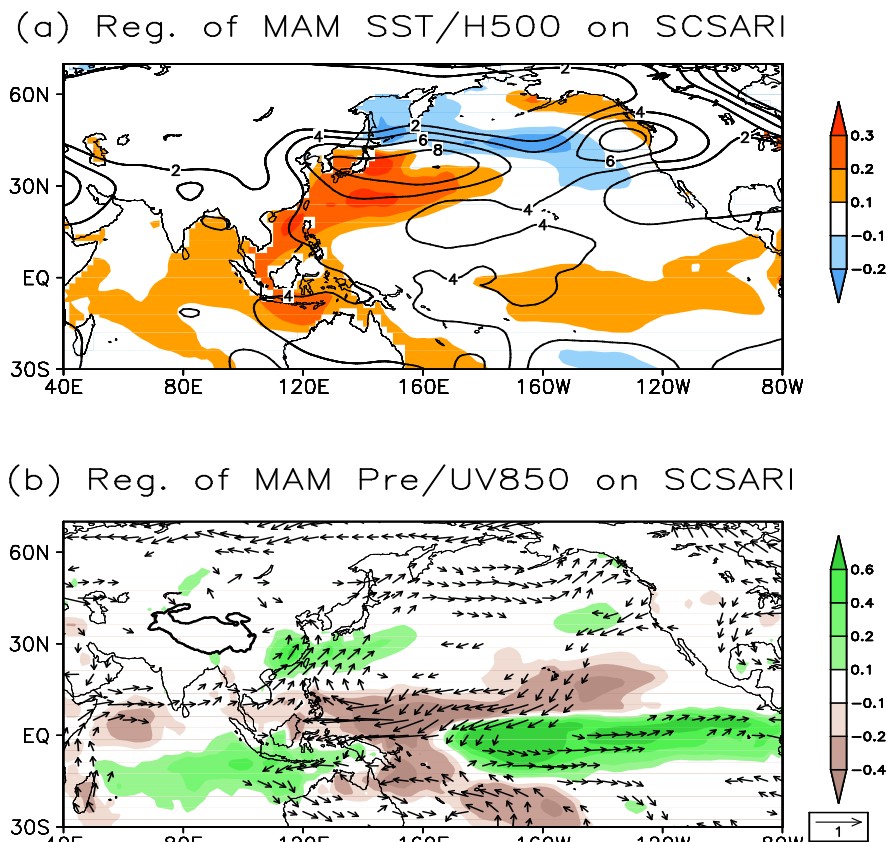

**Figure 3.** Regression of (**a**) SSTA (shading, exceeding the significance test of $\alpha = 0.05$, units: °C) and 500-hPa geopotential height anomalies (contour, exceeding the 0.05 significance level, units: gpm), (**b**) Spring precipitation anomalies (shading, exceed the significance test of $\alpha = 0.05$, units: mm/day) and 850-hPa wind anomalies (vector, both of the zonal and meridional winds exceeding the 0.1 significance level, units: m/s) with SCSARI.

The 500-hPa geopotential height anomalies associated with the SSTAs in the SCSAR show positive anomalies between the equatorial and mid-latitude regions. In the central Pacific, the 850-hPa wind field and precipitation anomalies (Figure 3b) indicate that anomalous northeasterly winds from the mid-latitudes strengthen the northeast equatorial trade winds. Anomalous anticyclone circulation occurs in the east of the Philippine Sea, which strengthens the subtropical high in spring. The sinking motion corresponding to the anticyclone causes less precipitation in the east of the Philippines. It is consistent with the conclusions obtained by Peng et al. [28] that when SSTA in the equatorial eastern Pacific Ocean is warmer in spring, an anomalous anticyclone appears in the SCS and the Philippines, and the subtropical high over the western Pacific is stronger and shifts westward. In the Indian Ocean, the cross-equatorial flow (the Somalian jet stream) intensifies, and anomalous westerly wind appears over the northern Indian Ocean and the Bay of Bengal. The anomalous southwesterly wind over the SCS converges with the southerly wind, causing wind convergence over the East China Sea and the Sea to southern Japan.

Meanwhile, the anomalous southwesterly winds bring sufficient water vapor from the SCS and the Philippines Sea, which leads to anomalous precipitation in South China, the Meiyu area, and the Sea to southern Japan. There is a strong positive precipitation anomaly in the eastern and central equatorial Pacific. On its northern side, the negative SSTAs in the area lead to less precipitation. When the MAM SSTAs in the SCSAR show a uniformly warming pattern, the precipitation in the Meiyu area is above normal, while the southern SCS presents less precipitation due to the divergence of wind.

Figure 4 shows the positive correlation between SSTAs and SCSARI in the Northwest Pacific was established in February and maintained to May. From February to April, the relationship between SSTAs in the eastern and central equatorial Pacific (the northern Indian Ocean) and warming in the SCSAR weakens gradually. We can find that the SCSAR MAM SSTAs are closely related to the MAM SSTAs in the Indian Ocean and the Pacific Ocean. The SSTA distribution in the Indian Ocean shows a uniformly warming, and the central equatorial Pacific Ocean is significantly warmer than normal. These two types of SSTA distribution pattern can be characterized by the Indian Ocean basin warming mode (IOBW, the average SSTA anomaly in the region of 20° S–20° N/40°–110° E) and Nino 3.4 index (the average SSTA anomaly in the eastern and central equatorial Pacific 5° N–5° S/170°–120° W). To reveal the correlations of SSTAs in the SCSAR with those in the Indian Ocean and the Pacific Ocean, we further examine the correlation of SCSARI with the IOBW and Nino 3.4 indexes at various leads and lags (Table 1). The long-term trend of the IOBW index has been removed.

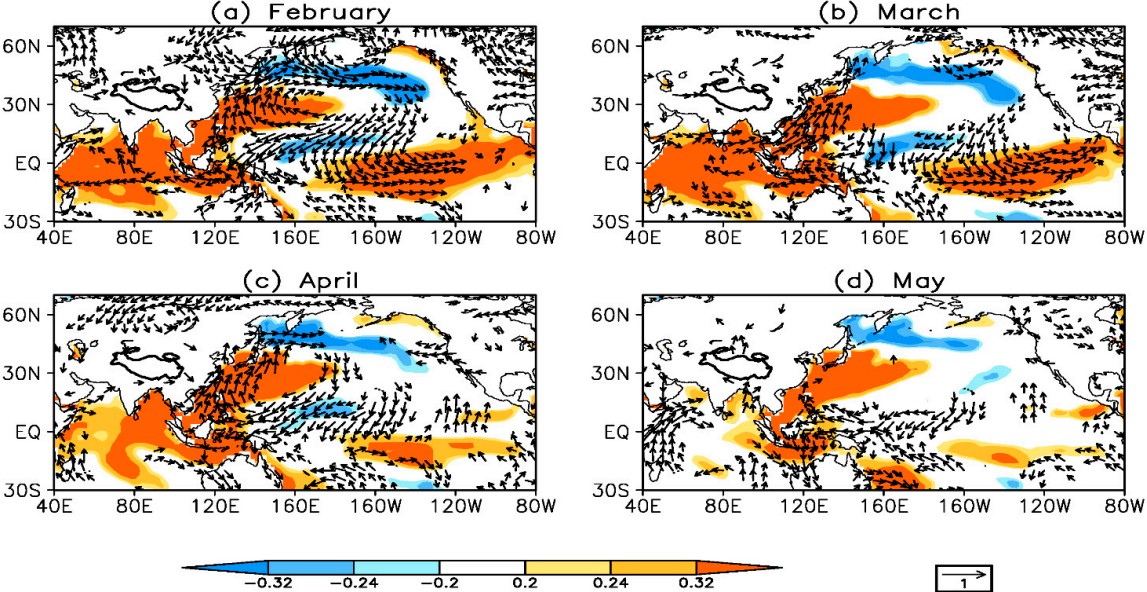

**Figure 4.** Spatial distributions of correlation coefficients between the SCSARI and SSTA (shading, exceeding the 0.05 significance level) and 850-hPa wind anomalies (vector, both of the zonal and meridional winds exceeding the 0.1 significance level). (**a**) February, (**b**) March, (**c**) April, and (**d**) May.

**Table 1.** Lead-lag correlation coefficients of SCSAR (South China Sea and its adjacent regions) Index with the IOBW (Indian Ocean basin warming mode) and the Nino 3.4. The left column represents the seasons of IOBW and Nino3.4. Values that exceed the 0.05 significance level are shown in bold.

| Lead-Lag Correlation | SCSAR Index (MAM (0)) | |
| :---: | :---: | :---: |
| | **IOBW** | **Nino3.4** |
| MAM (−1) | −0.02 | 0.07 |
| JJA (−1) | 0.04 | **0.41** |
| SON (−1) | **0.37** | **0.48** |
| DJF (−1) | **0.59** | **0.38** |
| MAM (0) | **0.58** | 0.19 |
| JJA (0) | 0.14 | −0.21 |
| SON (0) | 0.01 | **−0.25** |
| DJF (0) | −0.14 | **−0.28** |

Table 1 shows that the simultaneous correlation between MAM SSTAs in SCSAR and IOBW is 0.58, which is nearly equal to the maximum correlation coefficient of 0.59 when the SCSARI lags the Indian Ocean basin warming mode by three months. The strongest correlation between SCSARI and

Nino3.4 is 0.48 when the SCSARI lags the Nino3.4 by six months. However, the relationship between Nino 3.4 and MAM SSTAs in SCSAR rapidly weakens after the peak of ENSO-related SST anomaly. Given that the IOBW is highly dependent on the El Nino events [29], the cross-correlation results in Table 1 suggest that the MAM SSTAs in the SCSAR are affected by the SSTAs in the Indian Ocean. When the MAM SSTAs in the SCSAR shows the uniformly warming pattern, significant warming also occurs in most parts of the Indian Ocean.

The results above are confirmed by the spatial patterns of correlations of seasonal SSTAs with SCSARI (Figure 5). Figure 5a shows that there are significant positive correlations between the SCSARI and DJF SSTAs in the eastern and central equatorial Pacific Ocean and the basin-wide Indian Ocean. During spring, the positive correlation occurs in the Northwest Pacific and the eastern Indian Ocean (Figure 5b). However, according to Figure 5c–e, the signal of MAM SSTAs in the SCSAR is maintained to JJA but has weak lasting influences on SSTAs in other regions.

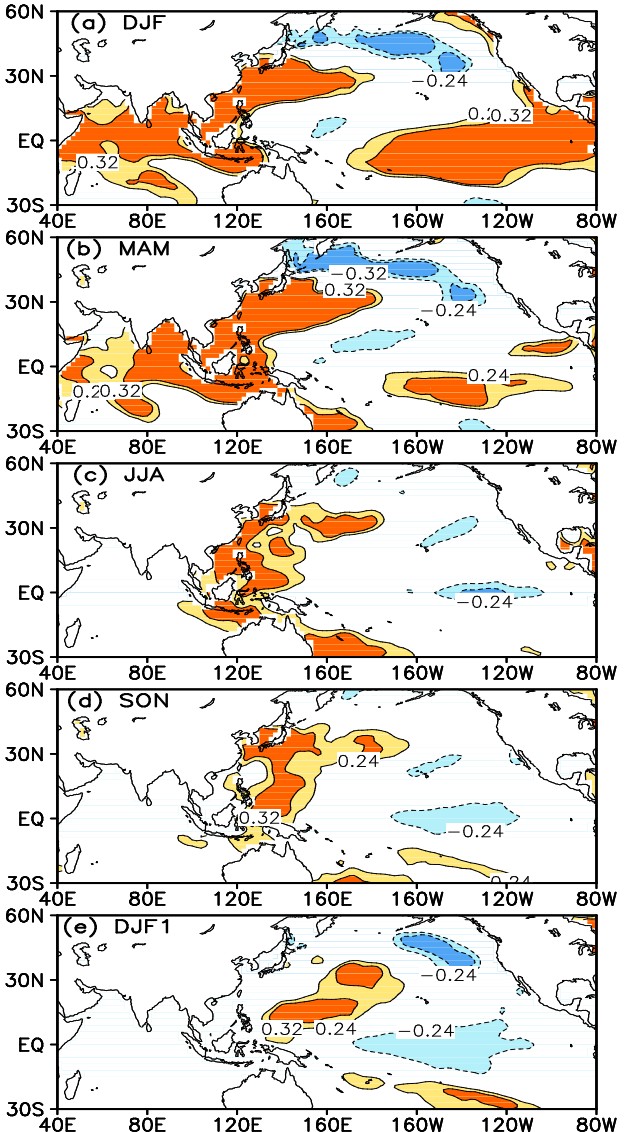

**Figure 5.** Spatial distributions of correlation coefficients between SCSARI and seasonal mean SSTAs in (**a**) preceding winter, (**b**) spring, (**c**) summer, (**d**) autumn, and (**e**) winter.

To further discuss the relationship between MAM SSTAs in the SCSAR and the Pacific Ocean, we use the method proposed by Frankignoul and Sennéchael [30] to define an index, namely SCSARI_IOBW. SCSARI_IOBW is defined as the standardized series of the MAM SSTAs in the SCSAR without the effect

of the detrended IOBW index. The regression analyses are carried out between the SCSARI_IOBW index and MAM SSTAs, 500-hPa geopotential height, rainfall, and winds at 850-hPa (Figure 6).

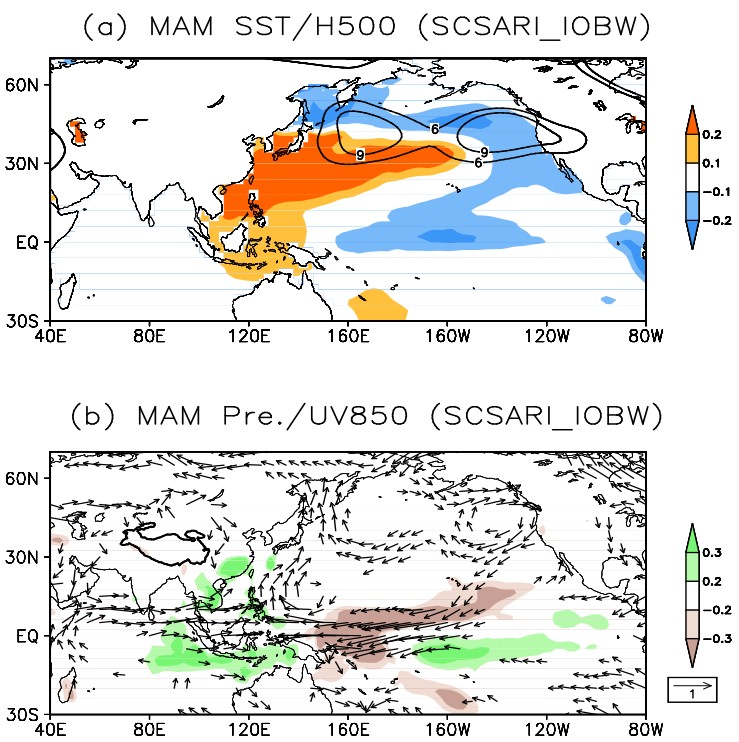

**Figure 6.** Same as Figure 3, but for the SCSARI_IOBW index, (**a**) SSTA and 500-hPa geopotential height anomalies, (**b**) Spring precipitation anomalies and 850-hPa anomalous wind.

It can be seen from Figure 6a that after removing the signal of IOBW from the SCSARI, MAM SSTAs in the SCSAR show larger value in the northern part but the relatively small value in the southern part. The positive SSTAs in the equatorial central and eastern Pacific shown in Figure 3a also disappear, but the SSTAs in the Northwest Pacific are still significant. In the Northwest Pacific, the intensity of the eastward extended "warm tongue" and the cold-water belt surrounding the "warm tongue" are both strengthened, and a strong cold center appears north of the equatorial central Pacific Ocean. The weakened SSTAs in SCSAR and intensified SSTAs in the Northwest Pacific indicates that MAM SSTAs in the northern part of SCSAR is more related to the SSTAs in the Northwest Pacific, while MAM SSTAs in the southern part of SCSAR has a closer relationship with the SSTAs in the Indian Ocean.

In the 500-hPa geopotential height field, the positive geopotential height anomalies at the low-latitudes in Figure 3a disappears, but two positive anomalies centers appear near 30° N. In the 850-hPa wind field, both the Somali cross-equatorial flow and the westerly wind in the northern Indian Ocean are strengthened (Figure 6b), indicating that the MAM SSTA warming in the SCSAR may strengthen the southern Asian monsoon in late spring. The intensity of the Philippine anticyclone over the western equatorial Pacific weakens and shifts eastward, resulting in a weaker and eastward-moved subtropical high in the western Pacific in spring. The anomalous precipitation zone, which is caused by the convergence of the northern SCS's southwesterly wind and the anomalous northerly wind on the west side of the Philippine anticyclone, also shows negative value and eastward movement. Compared to Figure 3, after the removal of the IOBW signal, the warming in the spring SCSAR SSTAs makes anomalous southwest wind speed decrease in the SCS. The associated wind anomalies can only reach the northern SCS and South China, resulting in less water vapor transported to the north, the scattered anomalous precipitation zone, and less precipitation.

It should be noted that after removing the strong influence of the Indian Ocean, the MAM SSTAs in the SCSAR can still affect the spring climate in East Asia, resulting in more precipitation in South Asia, South China, and southern Japan. The MAM SSTAs also plays a key role in linking the western Pacific subtropical high, the Philippine anticyclone, and the anomalous westerly wind in the Bay of Bengal.

### 3.3. Analysis of the Characteristics and Formation Mechanisms of the Second most Dominant Pattern of Spring SCSAR SSTAs

The regression between the time series of the second EOF mode (PC2) and each physical field in spring (Figure 7a) shows that when PC2 is positive the SST in the eastern and central equatorial Pacific is above normal and the triangular cold-water belt along the equator appears on both sides. During the same period, the 500-hPa geopotential height field shows a wave pattern with both positive and negative geopotential height anomalies over the East Asia region. From low to higher latitudes, a positive geopotential height anomaly center appears over the south of the Japan Sea and the East China Sea (centered at 30° N, 140° E). A negative geopotential height center appears over the Okhotsk Sea (centered at 60° N, 165° E) as well. The distribution of this geopotential height anomaly along the coast of East Asia resembles that of the Pacific-Japan wave train [31]. The propagation of atmospheric Rossby wave in spring leads to the wave train distribution in the Northern Hemisphere similar to that in the "great circle" path of stationary Rossby waves, with high-pressure center over the eastern Japan Sea, low-pressure center over the Okhotsk Sea, high-pressure center over the northern part of North America, and extraordinary low pressure over Mexico [32]. At the same time, affected by the ENSO, the eastern and central equatorial Pacific region stimulates a teleconnection pattern similar to the Pacific-North American (PNA) teleconnection waves from the equator to high latitudes [33–36]. Specifically, the eastern and central equatorial Pacific region is the center of positive geopotential anomaly, the Northeast Pacific and southern North America are the centers of low pressure, and a high-pressure center appears in the northwest of North America.'

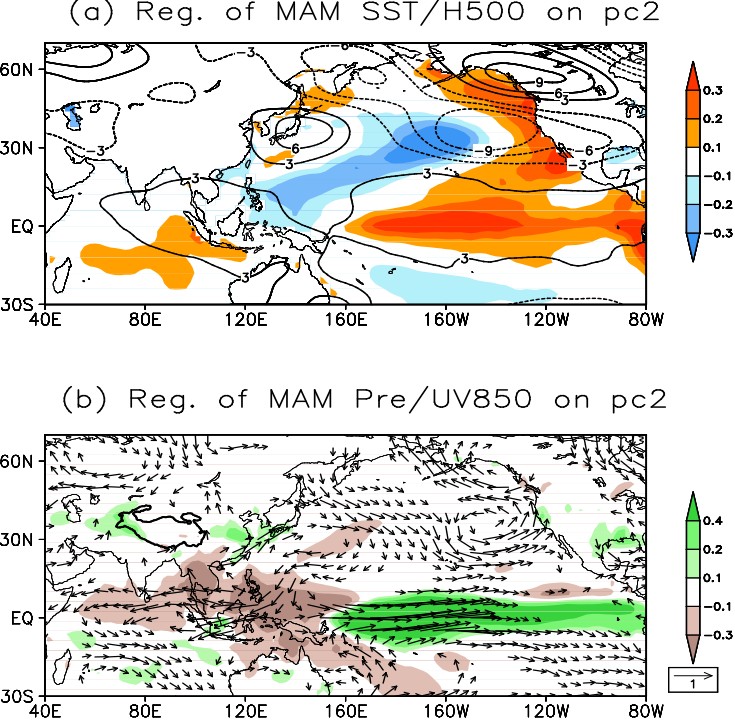

**Figure 7.** Same as Figure 3, but for PC2, (**a**) SSTA and 500-hPa geopotential height anomalies, (**b**) Spring precipitation anomalies and 850-hPa anomalous wind.

In the 850-hPa wind field (Figure 7b), an anomalous anticyclone circulation occurs over Southeast Asia, the SCS, and the eastern Philippine Sea. This circulation connects with the anticyclone circulation over the East China Sea and the Sea to the southern Japan. It makes the western-northwestern Pacific regions controlled by anomalous anticyclone circulation and the subtropical high shifts northward. The occurrence of anomalous anticyclone causes less precipitation over the Philippine Sea, the Marine Continent, and the SCS, which is consistent with the findings of Peng et al. [28]. They reported that when the MAM SSTA in the eastern equatorial Pacific Ocean was cooler (warmer) than normal, an anomalous cyclone (anticyclone) occurred near the SCS and the Philippines Sea, and the western Pacific subtropical high was weak (strong) and shifted eastward (westward). The northerly wind appears in the east of the anticyclonic circulation, and the anomalous northerly wind is split into two airflows in the western Pacific. One flows to the central equatorial Pacific Ocean to become anomalous westerly wind, which weakens the equator trade wind and inhibits the upwelling in the eastern Pacific. As a result, the El Niño signal with high SST in the eastern and central equatorial Pacific emerged. The other branch flows westward through the northern Indian Ocean and the Bay of Bengal, causing anomalous easterly wind in the northern Indian Ocean, which is integrated with the anomalous northerly wind in the western equatorial Indian Ocean, weakening the cross-equatorial flow off Somalia.

Thus, when the spring SCSAR SSTAs show sandwich pattern and the cold (warm) center is located in the east of the Philippines Sea, the SSTAs will cause anomalous anticyclone (cyclone) circulation over the SCSAR. This anticyclone (cyclone) circulation affects the atmospheric circulation over the Indian Ocean and the Pacific Ocean, which in turn affects the spring climate of the region.

To further reveal the energy sources and propagation paths of atmospheric wave activity, which is induced by precipitation anomalies and caused by the spring SCSAR SSTAs as seen in the second mode, we calculate the composite 500-hPa geopotential height field, the 200-hPa wave activity flux (TN flux), the stream function distribution by difference analysis, and the regression field of 200-hPa geopotential height. Figure 8 shows that positive anomaly of the 500-hPa geopotential height is significant at low latitudes, and there are two high-pressure centers: one located in the eastern equatorial Indian Ocean, the SCS and the Marine Continent, and the other located in the eastern and central equatorial Pacific. Along the East Asia region, there is a wave train with positive and negative geopotential height from low to high latitudes. The high- and low-pressure centers are consistent with the 500-hPa geopotential height distribution in Figure 7a.

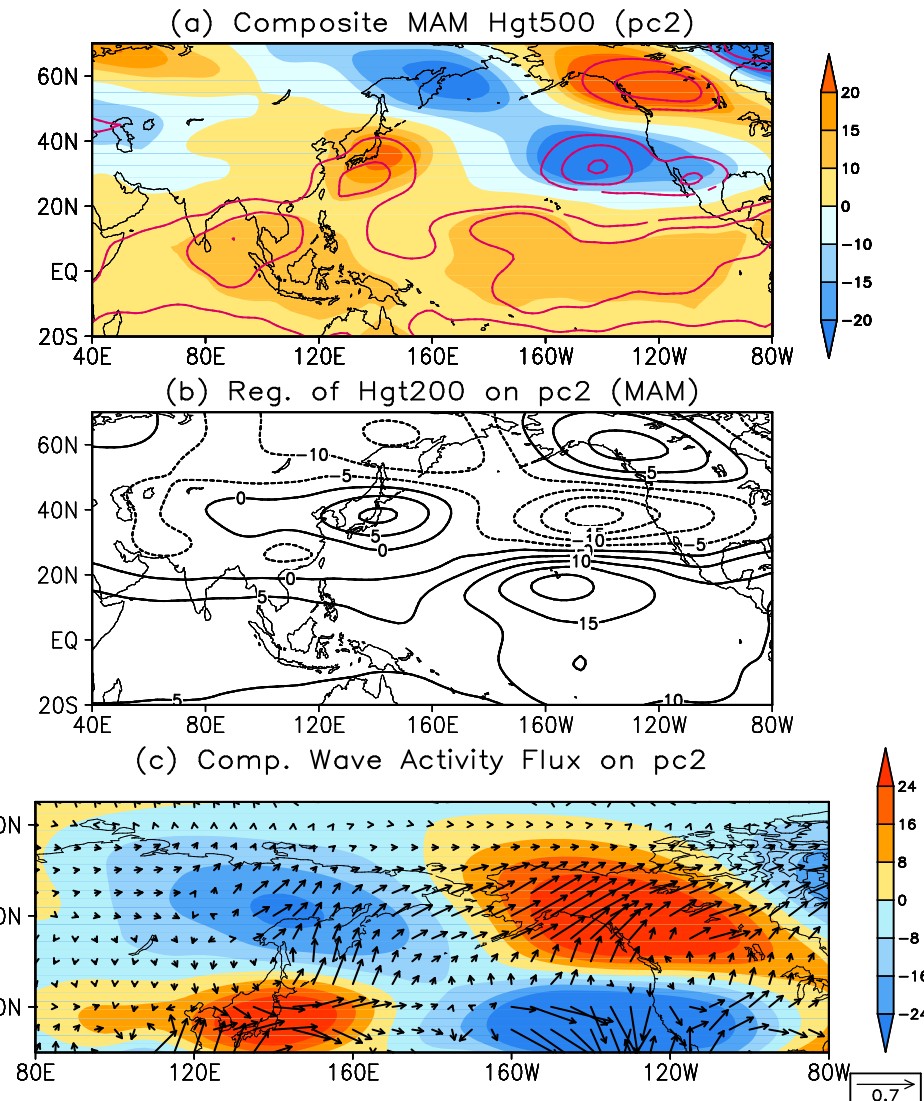

**Figure 8.** (**a**) Composite differences of 500-hPa geopotential height anomalies (units: gpm). Anomalies exceeding the 0.05 significance level are shown by red contours. (**b**) Regression of 200-hPa geopotential height anomalies (contour, interval = 3, gpm). (**c**) Composite differences of 200-hPa wave active flux (vector, units: m² s⁻²) and stream function (shading, units: m² s⁻¹).

The 200-hPa geopotential height regression is consistent with the 500-hPa composite diagram. It indicates that the atmospheric anomaly corresponding to the second mode has barotropic structures. In the eastern and central equatorial Pacific, there is a positive geopotential height anomaly symmetrical about the equator. Zheng et al. [37] pointed out that the equatorial symmetrical air pressure distribution in the Pacific region might be the Rossby response to the SSTAs in the eastern and central equatorial Pacific in spring [38,39].

Spring precipitation anomalies in the SCS and the eastern Philippines can excite a wave train from low to high latitudes along the coast of East Asia [37]. We can see that when the precipitation over the northern SCSAR is above normal in spring, a wave train is emitted from the SCSAR to East Asia, northwest Pacific, and North America in spring. Along with reduced 200-hPa geopotential height (Hgt200) over South China (centered at 25° N, 110° E), enhanced Hgt200 over the northeast part of China/Korean peninsula/Japan (centered at 40° N, 140° E), reduced Hgt200 over Okhotsk Sea Region (centered at 65° N, 150° E) and enhanced Hgt200 over northern North America (centered at 60° N, 130° W). Figure 8c shows that along the East Asian coast, the wave train spreads the energy

from mid-latitudes to higher latitudes through atmospheric teleconnection, and can even influence the North American atmospheric circulation in spring.

We use OISSTv2 datasets with a resolution of $1° \times 1°$ to repeat the analysis in this study. The OISSTv2-based results are highly consistent with the results obtained from ERSSTv3 (figures not shown).

## 4. Discussion

A large number of statistical, diagnostic, and numerical experiments indicate that tropical SSTAs have an important impact on the anomalous changes in atmospheric circulation at low, middle, and higher latitudes in Asia. The anomalous changes in the East Asian monsoon circulation in winter and summer are responses to the thermal conditions in the eastern equatorial Pacific. Previous studies have focused on the possible effects of winter and MAM SSTAs in the tropical Pacific and the Indian Ocean on the East Asian summer precipitation. We find in this study that the impacts of the MAM SSTAs in the SCS on the spring climate of East Asia and its neighboring areas cannot be ignored. The MAM SSTAs of the SCSAR play a key role in linking the western Pacific subtropical high, the Philippine anticyclone, and the anomalous westerly wind in the Bay of Bengal. It allows the airflow to converge in the area and modulating the East Asian climate. We also find that the MAM SSTAs in the SCSAR is not dominated by the ENSO, and only the second EOF mode exhibits a good correlation with ENSO signals. We only discuss the main modes and basic characteristics of the spring SCSAR SSTAs and speculate for the causes of the formation of different EOF modes of SSTAs. However, we have not discussed the causes of different SSTA modes in depth and the impact of spring SCSAR SSTAs on the summer climate thoroughly, which deserves further investigations.

## 5. Conclusions

By using multiple reanalysis data and applying EOF analysis, we explored the characteristics and formation mechanisms of spring SSTAs in the SCSAR. There are two modes of MAM SSTAs in the SCSAR. The first mode is the uniformly warming of MAM SSTAs, and the second mode shows a sandwich pattern with cold SSTA over the central SCSAR centered near $10°$ N, flanked by warm SSTA over the northern oceans near $25°$ N and in the subtropics near $10°$ S.

The spring SCSAR SSTAs are affected by the SSTAs in the Indian Ocean. The basin-wide warming in the Indian Ocean leads the MAM uniformly warming in SCSAR by 3 months, with the maximum correlation coefficient of 0.59. Related to the first EOF mode, above-normal precipitation is found in the Meiyu area, while the southern part of the South China Sea has less precipitation due to the divergence of wind.

For the second EOF mode, the anomalous anticyclone over Southeast Asia, the SCS, and the eastern Philippine Sea are associated with the sandwich SSTA pattern and above-normal precipitation in the northern part of SCSAR. The sandwich pattern causes less rainfall in the Philippine Sea, the Marine Continent, and the South China Sea. There is a wave train emitted from the SCSAR to East Asia, northwest Pacific and North America in spring, with negative center of geopotential height anomaly over South China, positive center over the northeast part of China/Korean peninsula/Japan, negative center over Okhotsk Sea Region and positive center over the northern North America. The wave train spreads the energy from mid-latitudes to high latitudes through atmospheric teleconnection, and can even influence the North American atmospheric circulation in spring.

**Author Contributions:** Conceptualization, X.H.; Methodology, W.G.; Writing—original draft preparation, W.G.; Writing—review and editing, W.G., S.Y., X.H., W.W., and Y.X.

**Funding:** This study was funded by the key project of Hunan Meteorological Administration (Grant XQKJ18A004), the general project of Hunan Meteorological Administration (Grant XQKJ19B050), the National Natural Science Foundation of China (Grant 41975074, 91637208 and 41661144019), the Natural Science Foundation of Guangdong Province (Grant 2017A030310571), the Fundamental Research Funds for the Central Universities (Grant 17lgpy21), the Zhuhai Joint Innovative Center for Climate-Environment-Ecosystem and the Jiangsu Collaborative Innovation Center for Climate Change, China.

**Acknowledgments:** We are grateful for the constructive and insightful comments from the editor and four anonymous reviewers that have led to a significant improvement in the presentation. The authors acknowledge the support from Chengyang Zhang and Yana Li for the generation of this manuscript.

**Conflicts of Interest:** The authors declare no conflict of interest.

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
