# Peer review of "Characteristics and Formation Mechanisms of Spring SST Anomalies in the South China Sea and Its Adjacent Regions"

_atmosphere, doi:10.3390/atmos10110649_

Round 1
Reviewer 1 Report
SUMMARY
In Characteristics of spring SST anomalies in the South China Sea and its adjacent regions and their possible impact on Asian climate, Gao et al. present EOF, regression, and correlation analyses of boreal spring sea surface temperature anomalies from the western subtropical Pacific and evaluate their role in modulating Asian precipitation. I recommend the paper could be published following revision and incorporation of the comments provided below.
SPECIFIC COMMENTS
Title. Consider removing the word possible from the title. It would seem improbable that SCS SSTAs do not impact Asian climate and your paper exactly aims to show that.
Line 22. Maybe there is a better way to describe this pattern rather than referring to it as a sandwich pattern?
Line 58. There could be a bit more of a set-up or motivation for why you focus on the spring SSTAs because the abstract seems to indicate that we are headed towards evaluation of how these spring SSTAs influence precipitation somewhere.
Line 112. Delete etc and be explicit about the analyses you completed
Line 118. Rather than omit the figure, zoom out a bit in fig 1 with a box around your area of interest.
Line 123. How were the sstas calculated? These are anomalies of mam ssts compared with the long term mean? And if so, the mean from what period?
Line 139. Consider replacing sandwich pattern with something less informal.
Line 167. Be careful with these descriptions. Thus far you have a sandwich, a warm tongue, a wedge, and a flipped C.
Line 192 and 196. Here you refer to southerly and northerly transport of water driven by the winds Be clear on whether you’re referring to water vapor or ssts.
Line 237. There is no 0.17 in Table 1. To what are you referring here or is this typo?
Line 331. Could you label some of these features or give coordinates for the reader, eg Okhotsk Sea.
Line 396. Define what is meant by H200 and be consistent with figs. See fig 8 comment.
Line 404. This first sentence is circular and unnecessary
Line 423. Uniform not uniformly. And these are not types as much as they are two leading eof patterns.
FIGURES
Fig. 1. It would be helpful to have a) zoomed out with a box around your SCSAR region so we could see how the regression pattern is distinct for the SCSAR region. Also, it is not clear what is color/contoured in a), the caption says anomalies, but anomalies compared with what? Is it the regression coefficient of the time-series in b) on the MAM SST time-series at each grid-point?
Fig 8. An earlier figure had a label of H500 and here you have hgt200. Be consistent about these labels.
Reviewer 2 Report
See attached.

Reviewer 3 Report
See attached.

Reviewer 4 Report
In this study, the authors revisit the relationship between the seas surface temperature variability in the boreal spring and the East Asian summer monsoon using empirical evidence. The manuscript may be publishable after a major revision.
EOF1 looks very similar to the spring climatology of SSTAs, an indication that annual and seasonal climatologies have not been removed when computing the anomalies. The authors need to redo the analysis using the correct anomalies. For a comprehensive description of methods adopted by the community for studying the East Asian monsoon the authors may refer to “East Asian Monsoon” edited by C. -P. Chang. A detailed description on how anomalies were computed should also be added to the manuscript.
Reviewer 5 Report
This paper can be substantially improved if physical explanations are clearly provided, which can be improved in part by proofreading service from language professional. For example, correlation analysis is the major approach in this study and the correlation maps between PSCSI and some fields (MAM SSTA, rainfall, winds at 850hPa, geopotential height at 500 hPa, etc) are used to explain the effect of the two dominant modes of SSTAs in the PSCS on SSTAs in other regions such as Northwest Pacific and/or the formation of these two dominant modes by SSTA in Indian Ocean and ENSO. These maps demonstrate the correlation and show some kind of temporal synchronization of these fields with the two dominant PCs of EOF, not the actual fields. For example, vector plot in Figure 3b is not the actual wind fields, but representing the strength and sign of correlation between winds and the PSCSI. I suggest to rewrite them. In addition, the paper is somewhat redundant and can be shortened.
Some minor comments listed below for consideration:
Line 20 - 21: other observational data --> please specify these observational data
Line 45: the region ---> please specify it
Line 47: other regions ---> not clear
Line 51: variations --> variability
Line 55 - 56: The SSTAs in the eastern ...and the Northwest Pacific. --> it needs a reference or references to support the statement
Line 72 - 74: When the SSTAs ... in North and South China. ---> it needs a reference or references to support the statement
Line 76 - 77: This study is to explore .... summer climate variations. It seems missing a name of region.
Line 106 - 107: Then, the spring SSTAs is calculated by the mean of MAM SSTAs.... ---> the spring SSTAs are defined as the mean of MAM SSTAs.
Line 115: onto --> both the sentence as well as the figure caption for Fig. 1 need revision. Other figure captions also need clear writing.
Line 142: various physical fields ---> please specify them.
Figure 1a: The map shading is not the actual temperature or SSTA, which is the correlation coefficient of SSTA. So a name called "warm tongue" for positive correlation needs justification.
Figure 2: EOF analysis, please specify the unit if appropriate.
Line 169: A preposition is missing between subtropical high and the western Pacific
Line 171 -174: The anomalous ..In this area. ---> Both sentences need revisions
Line 180 - 181: With the warm water ....is excessive. ---> not clear, please rewrite it.
Line 182: from ---> in
Table 1: please specify which leads/lags which
Line 252 - 253: Is it related to the topic of the present study?
Line 255: Please clearly define PSCSI_IOBW
Line 273 - 275: I am not sure such a strong conclusion can be made from a. correlation analysis.
Line 293: "southwest wind speed decrease" --> What I see from Figure 6b is the correlation between winds and PSCISI_IOBW gets weak.
Line 357 - 358: Fig. 8 shows the 500-hPa geopotential height is significantly positive ...---> positive anomaly of the 500-hPa geopotential height is significant at low latitudes.
Reviewer 6 Report
This paper investigated the feature of the spring SST over SCS and surrounding region, and the formation mechanisms.
The paper is written in a rather sloppy manner, the evidences cannot fully support the conclusion. Therefore, I think this paper cannot be accepted unless after major revisions. The following are my specific comments.
Two major comments:
For the first mode of SCS SST, since the SCS SST is highly coupled with the local wind and precipitation, this is a chicken-eggs thing on the causality between the air and sea. Any evidences for whether the SSTA in SCS is a resultant of the wind or the forcing of the wind? For the second mode of SCS SST, how could you state that the “precipitation anomalies trigger train emitted from the PSCS to East Asia, northwest Pacific and North America in spring” without removing the effect of ENSO? What is the underlying mechanism for the teleconnection?
Specific comments:
The name of Pan-SCS in the title, also in whole manuscript is quite misleading. Based on box in the Figure 1, the target domain of the study includes East China Sea, most of the Maritime continent and the entire SCS. Pan-SCS is not suitable to represent this domain. It is just the SCS and its surrounding regions. As you shown in the title of Figure 2, SSTA is detrended before EOF. Why you detrend the data? Since there is no significant trend as shown in Figure 1b. Did you already detrended in Figure 1b? Line 169, please rewrite this sentence. Line 177, southern Japan Sea? Or Sea to the southern Japan? Line 178, mid-higher latitudes means middle of the higher latitudes, if you want to express middle and high latitudes, use mid-to-high latitudes. Transporting warm water from the northern Indian Ocean to the SCS? How? Some references are not relevant at all. For example, none of the references from 9 to 12 is talking about the relationship between SCS SST and ENSO.
Round 2
Reviewer 2 Report
It is quite unclear to me whether the things the authors said they would change were changed. For example, they still talk about the use of the ERSSTv3 dataset; in the of the PREC dataset, the reference to it is completely left off; as are the citations for other datasets; e.g., Chen, M., P. Xie, J. E. Janowiak, and P. A. Arkin, 2002: Global Land Precipitation: A 50-yr Monthly Analysis Based on Gauge Observations, J. of Hydrometeorology, 3, 249-266
As for section 2.2, perhaps the authors mis-understood me when they say, "We have deleted the superfluous description in Lines 91-101"; so let me be clear, ALL of section 2.2 is superfluous; why show a three line equation for TN, when that variable is mentioned in passing once?
I continue to see a lack of a cogent paper that I can come away with any real addition to the literature.
Reviewer 4 Report
The authors have addressed my concerns. I recommend publication of the manusript as it is.
Author Response
Thanks for your constructive suggestions that have led to a significant improvement in the presentation. We also seek editing help from the native speaker to improve the language.
Reviewer 6 Report
The authors have addressed all my concerns. Therefore, I think it can be accepted by Atmosphere.
Round 3
Reviewer 2 Report
Okay, I understand using ERSST with OISST as only a verification, but then why not use the latest ERSST v5. Also, by the way (line 93 or so), OISST has a resolution of 0.25x0.25 degrees.
Author Response
Thanks for your comment. The resolution of OISSTv2 is 1°×1°, which is not a mistake. Moreover, for our study of inter-annual variability, resolution of 1°×1° is adequate. As the information of any higher resolution that would not be based on observations (we don’t have 0.25 by 0.25 degree observation network globally at daily time scale for anything (even for satellite observations). The ultra-high resolution data is based on mathematical interpolation, which is intended to be used for short-term forecasting using high-resolution numerical models.